# QTc Intervals Are Prolonged in Late Preterm and Term Neonates during Therapeutic Hypothermia but Normalize Afterwards

**DOI:** 10.3390/children8121153

**Published:** 2021-12-08

**Authors:** Karel Allegaert, Thomas Salaets, Robert M. Ward, Pieter Annaert, Anne Smits

**Affiliations:** 1Department of Development and Regeneration, KU Leuven, Herestraat 49, 3000 Leuven, Belgium; anne.smits@uzleuven.be; 2Department of Pharmaceutical and Pharmacological Sciences, KU Leuven, Herestraat 49, 3000 Leuven, Belgium; pieter.annaert@kuleuven.be; 3Department of Clinical Pharmacy, Erasmus MC, Postbus 2040, 3000 GA Rotterdam, The Netherlands; 4Division of Pediatric Cardiology, Department of Pediatrics, University Hospitals Leuven, 3000 Leuven, Belgium; thomas.1.salaets@uzleuven.be; 5Division of Neonatology and Clinical Pharmacology, Department of Pediatrics, University of Utah, Salt Lake City, UT 84132, USA; Robert.Ward@hsc.utah.edu; 6Neonatal Intensive Care Unit, University Hospitals Leuven, Herestraat 49, 3000 Leuven, Belgium

**Keywords:** therapeutic hypothermia, newborn, QTc interval, QTc prolongation, pharmacovigilance

## Abstract

Background: There are anecdotal reports on reversible QTc prolongation during therapeutic hypothermia (TH) for moderate to severe neonatal encephalopathy after asphyxia. As the QTc interval is a relevant biomarker for pharmacovigilance during medication development, a structured search and review on published neonatal QTc values to generate reference values is warranted to facilate medication development in this specific population. Methods: A structured search and literature assessment (PubMed, Embase, and Google Scholar) with ‘Newborn/Infant, QT and hypothermia’ was conducted (October 2021). Retrieved individual values were converted to QTc (Bazett) over postnatal age (day 1–7). Results: We retrieved 94 QTc intervals (during TH (n = 50, until day 3) or subsequent normothermia (n = 44, day 4–7)) in 33 neonates from 6 publications. The median (range) of QTc intervals during TH was 508 (430–678), and 410 (317–540) ms afterwards (difference 98 ms, or +28 ms/°C decrease). Four additional cohorts (without individual QTc intervals) confirmed the pattern and magnitude of the effect of body temperature on the QTc interval. Conclusions: We highlighted a relevant non-maturational covariate (°C dependent TH) and generated reference values for the QTc interval in this specific neonatal subpopulation. This knowledge on QTc during TH should be considered and integrated in neonatal medication development.

## 1. Introduction

Therapeutic hypothermia is the standard intervention for late preterm and term neonates with moderate-to-severe hypoxic-ischemic encephalopathy. Although effective, there is still a relevant burden of neurologic impairment in nearly 50% of treated infants. There is an active search for interventions to add to therapeutic hypothermia to further improve outcome [1,2,3,4,5]. As therapeutic hypothermia and asphyxia affect neonatal physiology, this will have impact on population-specific pharmacovigilance [3,4,5]. QTc (c = corrected, for heart rate) prolongation is crucial in pharmacovigilance, as a biomarker for ventricular repolarization disturbances and risk for torsade de pointes. In the absence of specific recommendations in neonates, guidance suggests the use of partial extrapolation from data in adults and older children. This is because event and intervention are believed to be similar in pediatric patients—including neonates—and adults, while exposure-relationship and safety are inadequately defined or thought not to be sufficiently similar [4,6,7].

The latest meta-analysis reports that therapeutic hypothermia alters QTc and may affect cardiac safety [1]. This has also been reported in randomized controlled trials [8,9,10]. Sinus bradycardia (<80 beats per minute (bpm), was more common (effect size 1.59, 95% CI 4.94–27.17) during therapeutic hypothermia. However, the incidence (0.6%) of major arrhythmia was similar between hypothermia cases (n = 2/406) and controls (n = 3/410), without quantification of ‘any arrhythmia’ or ‘prolonged QT’ intervals [1]. The National Institute of Child Health and Human Development (NICHD) (2000–2003) study reported one ‘persistent bradycardic event’ in the therapeutic hypothermia group, and two ventricular tachycardia events (one in the hypothermia and one in the normothermia group) [1]. In the CoolCap (1999–2002) study, major cardiac arrhythmia events were not observed [1]. In the Infant Cooling Evaluation (ICE 2001–2007) trial, prolonged QT (definition >98th centile for heart rate and age was observed in 31 (43%) of therapeutic hypothermia-treated cases compared to 19.7% in control asphyxia cases [8], but no arrhythmias required intervention [9]. Finally, in the Neo.nEURO study, cardiac arrhythmias were observed in 3/62 therapeutic hypothermia and 4/63 normothermia cases [10].

Consequently, the latest meta-analysis and randomized controlled trials are reassuring if we focus on the incidence (0.6%) of major arrhythmia that requires intervention in this high-risk population. However, this analysis does not provide information on QTc interval changes during and after therapeutic hypothermia. This is a relevant biomarker to assess the benefit/risk balance of pharmacotherapy added to therapeutic hypothermia [3].

Besides its relevance for medication safety, there is also diagnostic relevance. In a recent case report in this journal, the postnatal management of a newborn with a congenital long QT syndrome was described [11]. The newborn had ventricular fibrillation at birth, and subsequently underwent therapeutic hypothermia [11]. In such a scenario, one needs to compare QTc intervals collected during therapeutic hypothermia to relevant reference values.

We therefore conducted a structured search on published individual QTc intervals and QTc patterns in cohorts of neonates undergoing therapeutic hypothermia and afterwards during normothermia to construct a QTc pattern over postnatal age during and after therapeutic hypothermia.

## 2. Materials and Methods

A structured search was performed on 29 October 2021 in PubMed, Embase, and Google Scholar, with the search terms ‘Newborn OR Infant’, and ‘hypothermia AND QT’, without limitations on publication date or language. All hits were screened on title and abstract by the first author (K.A.). If judged to be relevant (inclusion: QTc data reported, in human neonates undergoing therapeutic hypothermia or shortly afterwards, exclusion: other topics, like unintentional hypothermia), the full paper was read. If subsequently retained for analysis, references and citations (backward and forward snowballing) were further checked for other relevant papers. The clinical studies retained in the latest meta-analysis on therapeutic hypothermia were also screened, using the same approach of reference and citation screening [1].

Individual QTc data reported in the papers were extracted, either paired (during and after hypothermia in the same patient) or unpaired (during and after hypothermia, in different patients). When raw data were provided in figures, individual observations were extracted (WebPlot Digitizer, Ankit Rohatgi, available online: https://automeris.io/WebPlotDigitizer/, accessed on 29 October 2021) [12]. When QT and heart rate were reported, data were converted to QTc (Bazett) based on the formula (QTc (ms) = QT/√RR interval). Statistics were descriptive (median and range, or mean and standard deviation). QTc (Bazett) during and after therapeutic hypothermia were compared (Mann–Whitney U test), and trends over postnatal age in both time intervals (day 1–3: therapeutic hypothermia or day 4–7: normothermia) were explored by correlation analysis. A *p*-value < 0.05 was considered statically significant (MedCalc^®^, Ostend, Belgium).

Papers reporting on QTc patterns in cohorts during and after therapeutic hypothermia, but without individual QTc data were assessed to confirm or adapt the QTc pattern constructed from the pooled individual QTc data in the current review.

## 3. Results

### 3.1. Search Results

The search in PubMed on ‘Newborn or Infant, hypothermia and QT’ resulted in six and nine hits (newborn and infant, respectively). The Embase search with the same words resulted in 22 and 22 hits. A Google Scholar search (including citations) resulted in 11,200 and 12,200 hits. The first 500 hits (listing strategy: priority) were screened on relevance, but provided no additional information. Six papers on individual QTc observations, and four additional cohorts describing QTc patterns over postnatal age were retained.

### 3.2. Individual QTc Observations

Based on six papers, we retrieved individual QTc observations during therapeutic hypothermia (n = 50, until day 3) and afterwards (n = 44, day 4–7) in 33 individual neonates [13,14,15,16,17,18]. The largest dataset contained 37 and 39 QTc intervals during and after therapeutic hypothermia in 19 neonates [13]. A second dataset reported on 10 QTc intervals in 10 cases only during therapeutic hypothermia [14]. The data from both cohorts were further extended by four individual cases (five QTc intervals during therapeutic hypothermia, three afterwards) [15,16,17,18]. The data did not allow further analysis of other potentially relevant covariates like gestational age, growth restriction, pH at birth, moderate or severe encephalopathy, survival, or biochemistry. We were unable to pair (except for the single case reports [15,16,17,18]) consecutive QTc measurements.

In almost all cases, the QTc prolongations were documented during clinical care, and without clinical interventions. However, in the Blengio case, the temperature was raised from 33.5 to 34.5 °C which shortened the QTc interval from 619 to 480 ms. Afterwards during normothermia, QTc was 317 ms. In the Farmeschi case of a late preterm neonate, (34 weeks, 6 days), the QTc interval during therapeutic hypothermia was 581 ms, and the clinicians decided to stop therapeutic hypothermia. Interestingly, before initiation of therapeutic hypothermia, the QTc time was 520–548 ms. No information on QTc during subsequent normothermia was provided.

Figure 1 provides the individual QTc intervals over postnatal age in the first week of life, with the first 3 days during therapeutic hypothermia (50 observations, 18 on day 1, 18 on day 2 and 14 on day 3) up to postnatal day 4–7, during normothermia (44 observations, 18 on day 4, 10 on day 5, 11 on day 6, and 5 on day 7) [13,14,15,16,17,18]. Based on these observations, the median (range) of the QTc interval during therapeutic hypothermia was 508 (430–678) ms (median QTc 522, 508 and 500 ms on day 1, day 2 and day 3 respectively). Afterwards (day 4–7), it was 410 (317–540) ms (median QTc 436, 412, 399 and 385 on day 4, day 5, day 6 and day 7, respectively). The difference between both settings is 98 ms (*p* < 0.001). Assuming a temperature difference between both settings of 3.5 °C, this is equal to +28 ms/°C decrease. During therapeutic hypothermia, 27/50 QTc intervals were >500 ms, of which 20/50 were >520 ms. Despite the consistent decrease in median QTc over postnatal age, the trends during therapeutic hypothermia (postnatal days 1–3) or afterwards (postnatal days 4–7) were not significant. A mixed model approach was impossible and only unpaired analysis was performed, as the published data did not allow to discriminate between paired and unpaired observations.

### 3.3. Cohorts

We also retrieved four reports on patterns of QTc changes over postnatal age during therapeutic hypothermia and afterwards. Cavallaro et al. compared differences in 31 neonatal cardiovascular parameters between moderate (33.5 °C) and deep hypothermia (31 °C). The authors observed that 2/14 and 3/17 had a QTc interval >520 ms during mild and deep hypothermia respectively [19]. Horan et al. quantified the mean QTc interval from 37 to 34 °C (431, 459, 445, and 465 ms at 37, 36, 35, and 34 °C respectively) in 27 neonates undergoing therapeutic hypothermia during extracorporeal membrane oxygenation (ECMO). They reported a 3.12 ms increase/°C decrease [20]. Lasky et al. also recorded the QTc interval repeatedly in two newborns with asphyxia during therapeutic hypothermia and subsequent rewarming. For each increment of 1 °C during rewarming, there was an increase of 9.2 bpm, and a QTc interval decrease by 21.6 ms/°C [21]. Finally, Montaldo et al. associated a longer QTc and lower heart rate during therapeutic hypothermia with better neurodevelopmental outcome at 18–24 months [22]. Using a paired data analysis, QTc intervals were measured on admission, during hypothermia (12, 24, 36, 48, 60, and 72 h) and subsequently during normothermia. The authors described a progressive increase in QTc intervals up to 36–48 h, with a subsequent decrease beginning during therapeutic hypothermia which normalized after rewarming [22].

## 4. Discussion

### 4.1. Findings

Using a structured search and pooling of reported individual QTc intervals, we constructed a pattern of reference values for QTc in late preterm and term neonates during therapeutic hypothermia and immediately afterwards (Figure 1) [13,14,15,16,17,18]. This pattern shows a significantly prolonged QTc interval during therapeutic hypothermia, with subsequent trend to normalization immediately afterwards. This pattern, based on individual measurements, was reported by six different groups and confirms the Montaldo cohort pattern that was based on rich sampling in a single cohort in one intensive care unit [22]. When quantified by incremental changes in temperature (/°C), this increase in QTc is ‘°C dependent’. Based on the differences in median QTc interval during therapeutic hypothermia and normothermia, it was estimated to be +28 ms/°C decrease, similar to the observation during rewarming in two cases (−21.6 ms/°C) [21]. In an ECMO setting and perhaps related to differences in hemodynamics, this was only 3.12 ms/°C [20]. From a pharmacovigilance perspective, it is important to notice that these large increases in QTc during therapeutic hypothermia are only very rarely associated with clinically relevant events, like ‘major arrhythmia’, except for sinus bradycardia.

Based on 94 individual QTc observations in 33 neonates during of following therapeutic hypothermia (day 1–7), the median QTc was significantly prolonged (504 versus 410 ms) during therapeutic hypothermia (day 1–3) compared to subsequent normothermia (day 4–7). This difference is equal to −21.6 ms/°C, where this pattern constructed from the individual findings is similar to the pattern described in four cohorts. It is hereby important to realize that this pattern of QTc intervals during and after therapeutic hypothermia is a phenotypic description, driven by hypothermia, but likely also in part determined by other covariates like potassium levels, pH, or disease severity.

### 4.2. Limitations

This pooled analysis obviously has limitations. These relate to maturational changes, potential publication bias, and the limitations of the Bazett correction in neonates that overestimates QTc at faster heart rates, which are typical of newborns.

Based on reports in healthy term neonates, a maturational decrease in QTc over the first days of postnatal age was expected [23,24]. Ulrich reported on maturational changes in QTc interval in 114 (pre) term neonates in the first week of life (range 31–≥37 weeks gestational age) [24]. From day 1 to day 4, QTc decreased (−10.3 ms/day) in the term newborn [24]. However, this maturational pattern was not statistically confirmed in our dataset, likely due to the limited number of observations, the limitations of the need to use unpaired analysis, and the dominant effect of therapeutic hypothermia (+28 ms/°C). Table 1 shows the median QTc over postnatal age, as reported by Ulrich and as documented in this pooled data analysis in neonates (highlighted in grey) during and after therapeutic hypothermia [13,14,15,16,17,18,24].

A publication bias due to ‘positive’ cases should also be considered. However, the pattern observed in the pooled individual cases was also observed in median trends and patterns in the reports of cohorts [13,14,15,16,17,18,19,20,21,22].

Finally, Bazett over-corrects at elevated heart rates and under-corrects at heart rates below 60 bpm, so that Fridericia’s correction is more accurate than Bazett in subjects with altered heart rates [25]. However, as we only had access to Bazett corrections in the publications, or QT with heart rate, we could report only pooled QTc Bazett data. We have to take into account that therapeutic hypothermia commonly results in relative bradycardia (<100 bpm), so that any difference related to Bazett or Fridericia correction will likely be limited.

### 4.3. How to Integrate These Findings in Pharmacovigilance Practices in Neonates

The pharmacovigilance practice to monitor for QTc effects during drug development was introduced after medication-induced arrhythmias were identified as the cause of quinidine-associated syncope [26]. Following this pivotal observation, an extensive list of medications that prolong the QTc interval emerged [27]. Regulations were developed for ‘thorough QT/QTc studies’ as part of drug development and approval. Subsequent revisions reflected on how to evaluate medication-induced changes in QT when a ‘thorough QT/QTc study in healthy adult volunteers’ cannot be conducted for safety related issues [25]. Exclusion of study subjects with a QTc longer than the usual adult standard (QTc 440 ms) would prevent enrollment of most newborns undergoing therapeutic hypothermia in clinical trials [3,4].

It is still a matter of debate about how ‘QT/QTc pharmacovigilance’ should be applied in neonates, with the combination of maturational and non-maturational changes in the QTc interval (Table 1) [13,14,15,16,17,18,24].

With this structured search, we highlighted a relevant non-maturational covariate which produces a °C-dependent prolongation of QTc, similar to the PD effect of therapeutic hypothermia on seizure control [28,29]. There are reports on QTc interval prolongation assessment for medications known to affect the QTc interval in adults during exposure in neonates. Cisapride prolonged the QTc interval in preterm and term neonates by +20 ms [30,31], while—in contrast to adults—domperidone showed no effect on the mean QTc interval [32]. It remains difficult to put this into perspective, especially for pharmacovigilance in the first week of life, because of maturational and non-maturational factors (Table 1) [13,14,15,16,17,18,24]. This is especially relevant to studies of medications to further improve neurodevelopmental outcome after asphyxia when added to therapeutic hypothermia.

In the absence of robust guidance, we suggest that in vitro (hERG) and in vivo (healthy adult volunteers) screening should be used to identify potential risks of QTc prolongation [33]. It was recently suggested that a double negative assessment be used as an effective screen for risks of QTc prolongation: negative in vitro human ether-a-go-go-related gene (hERG); and minimal, or no, in vivo QTc prolongation at concentrations exceeding clinical exposure. Medications that pass the double negative screen are associated with such a low probability of clinical QTc interval prolongation and medication-induced torsade de pointes that ‘double’ negative medications would subsequently not need detailed clinical QTc interval evaluation [33,34]. This approach is likely to be relevant for a medication development program focused in early neonatal life. In vivo phenotypic assessment in neonates will have major limitations in recognizing a potentially relevant signal in QTc because of the extensive inter- and intra-patient variability, which is even more pronounced during therapeutic hypothermia.

## 5. Conclusions

A pattern of QTc interval reference values in late preterm and term neonates during therapeutic hypothermia and immediately afterwards was constructed. This reflects a °C dependent, reversible effect on the QTc interval (+28 ms/°C decrease), with a median (range) of 508 (430–678) ms during TH. Consequently, we have highlighted that therapeutic hypothermia (°C dependent) is a relevant non-maturational covariate, in addition to the existing maturational QTc interval in early neonatal life. The knowledge of QTc interval variability should be considered and integrated into neonatal medication development and in studies of neonates undergoing therapeutic hypothermia.

## Figures and Tables

**Figure 1 children-08-01153-f001:**
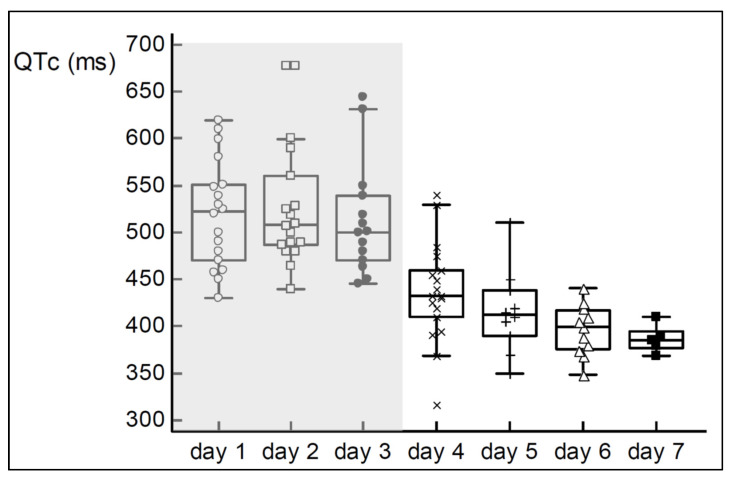
Pooled individual QTc intervals over postnatal age with 94 observations in 33 neonates (therapeutic hypothermia (light grey) day 1–3; normothermia (white), postnatal day 4–7), data from [13,14,15,16,17,18].

**Table 1 children-08-01153-t001:** Median QTCc values and the decrease in ms/day (for the healthy term, over 4 days (−10.3 ms/day) for the asphyxia cases over 3 days during therapeutic hypothermia (−11 ms/day), or between day 3 and day 4, from therapeutic hypothermia to normothermia, −54 ms) as reported in healthy term or asphyxia cases. Observations collected during hypothermia were highlighted in grey, data from [13,14,15,16,17,18,24].

Cohort	Day 1	Day 2	Day 3	Day 4	ms/Day
healthy term	444 ms	429 ms	421 ms	413 ms	−10.3 ms
asphyxia	522 ms	508 ms	500 ms	436 ms	−11 ms and −54 ms

## Data Availability

The raw data collected are available upon reasonable request and based on a protocol by the corresponding author.

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
