# Peer review of "QTc Intervals Are Prolonged in Late Preterm and Term Neonates during Therapeutic Hypothermia but Normalize Afterwards"

_children, 2021, doi:10.3390/children8121153_

Round 1

Reviewer 1 Report

Karel Allegaert et al. performed a structured search and review on published QTc values to generate reference values in infants with HIE undergoing therapeutic hypothermia. The work is of interest and can help clinicians in the management of these infants.

The following points can improve further the paper:

- Was the protocol registered with PROSPERO? Was the Cochrane handbook for systematic reviews of interventions used to frame this review?

- Who searched the literature?

- In the methods I noticed that the search was performed in PubMed, Google Scholar and Embase. What about Medline (Ovid) and Web of Science Core Collection (Web of Science, all citation indexes) databases Medline (Ovid)? Also the time frame of the search is not specified and the authors just state that a structured search was performed in October 2021.

-  The flow chart of the literature search is missing.

- The authors have not discussed the effect of the severity of HIE on the QTc interval increase. This is extremely important and can also have impact on the reference ranges the authors describe. In figure 1 I would set a different colour according to the different HIE stage and outcome if available.

Author Response

Karel Allegaert et al. performed a structured search and review on published QTc values to generate reference values in infants with HIE undergoing therapeutic hypothermia. The work is of interest and can help clinicians in the management of these infants.

We thank the reviewer for the overall positive assessment of our paper.

The following points can improve further the paper: Was the protocol registered with PROSPERO? Was the Cochrane handbook for systematic reviews of interventions used to frame this review?

We would like to stress that we never intended and have not reported that this is a systematic review (cf your summary: structured search). We intended to construct a QTc pattern over postnatal age during and after therapeutic hypothermia based on pooled analysis of individual QTc observations.  We also compared this pattern to results in other papers that described QTc patterns in cohorts of subjects over time. We have tried to describe this approach in the methods section, and have added subheading in the results section to make this clearer.

Who searched the literature?

This was already explicitly mentioned in the first version (K.A.) by the initials. We have added that this is also the first author of the paper in an attempt to make this even clearer.

In the methods I noticed that the search was performed in PubMed, Google Scholar and Embase. What about Medline (Ovid) and Web of Science Core Collection (Web of Science, all citation indexes) databases Medline (Ovid)? Also the time frame of the search is not specified and the authors just state that a structured search was performed in October 2021. The flow chart of the literature search is missing.

We have described our search strategy, the ‘hits’ collected and the papers retained, so that in our opinion this is better than using a flow chart as this would suggest a systematic review, and this is not what we have done. We hope that the reviewer will agree that we felt that it is scientifically more correct not to use tools commonly used in systematic reviews or meta-analysis that would be misleading. The date has been added.

The authors have not discussed the effect of the severity of HIE on the QTc interval increase. This is extremely important and can also have impact on the reference ranges the authors describe. In figure 1 I would set a different colour according to the different HIE stage and outcome if available

We agree that we also assume that other covariates (potassium, pH, disease severity) likely affect the QTc pattern, but we had no access to such data, so that we are not able to discriminate between the 50 QTc measures during hypothermia, or the 44 QTc measures after rewarming. We have retrieved data on QTcd (dispersion) in asphyxia cases (irrespective of cooling), but the results were driven by potassium and sodium (Zhao et al, Eur Rev Md Pharmacol Sci 2018), without additional difference between moderate or severe HIE cases. However, QTc-dispersion is another (less well understood) constructed ECG variable, so that we have not included this in the revised version

Reviewer 2 Report

The introduction section is very confusing. I suggest inserting only useful information in order to introduce the aims. Moreover, the aims should be clearly described. Please, improve this section.
The material and methods section is insufficiently described. The search strategy should be better described in order to allow repetition. Moreover, I suggest subdividing it into a subsection: it could be very useful for the readers. 
The result section should be improved. I suggest inserting tables summarizing the main data of each selected study. 
The discussion section should be improved. The authors should insert the strength and limitations of this study. 
Minor points: 
- Please, check the reference style, adjusting it in accord with the authors' guidelines.
- Please, check the English language: several phrases are hard to read.  

Author Response

Reviewer 2

The introduction section is very confusing. I suggest inserting only useful information in order to introduce the aims.

As also requested by the editor, a major part of the introduction has been removed to improve the text flow, to end with the aims of this study.

Moreover, the aims should be clearly described. Please, improve this section.

We have adapted this text section.

The material and methods section is insufficiently described. The search strategy should be better described in order to allow repetition. Moreover, I suggest subdividing it into a subsection: it could be very useful for the readers. 

Subsections have been added in the results and discussion section.  Different terms have been used in the methods section to stress both targets (patterns of individual QTc  and the patterns in a limited number of cohorts (n=4) retrieved).

The result section should be improved. I suggest inserting tables summarizing the main data of each selected study. 

The result section has been provided with subheadings, while the 6 papers retained for the individual QTc and the 4 cohorts have been described. We hereby hope that this has improved the results section in the opinion of the reviewers.

The discussion section should be improved. The authors should insert the strength and limitations of this study. 

We hope that the subheadings, and the removal of the section on animal experimental observations has improved the text flow and the ‘messages’ of the paper.

Please, check the reference style, adjusting it in accord with the authors' guidelines.

We thought that we respected the guidelines, but had added the doi as the journal provided an additional link (à), and we aimed to facilitate this process. We have removed these doi’s.

Please, check the English language: several phrases are hard to read

The authorship includes a native English speaking co-author, and he has re-checked the paper again on language. 

Round 2

Reviewer 1 Report

The paper has overall improved and the use of subheadings makes the paper much clearer.

I understand that the authors did not intend to conduct a systematic review, and they only wanted  to construct a QTc pattern over postnatal age during and after therapeutic hypothermia. However I think the aim can be better described in the introduction where the author state" We therefore conducted a structured search on published individual QTc intervals and QTc patterns in cohorts of neonates undergoing therapeutic hypothermia and afterwards to generate such reference values."

I suggest the authors add to the discussion that other covariates (potassium, pH, disease severity) likely affect the QTc pattern. However, they had no access to such data data. Therefore, they could not address this point in the paper. 

Author Response

The paper has overall improved and the use of subheadings makes the paper much clearer.

We thank the reviewer

I understand that the authors did not intend to conduct a systematic review, and they only wanted  to construct a QTc pattern over postnatal age during and after therapeutic hypothermia. However I think the aim can be better described in the introduction where the author state" We therefore conducted a structured search on published individual QTc intervals and QTc patterns in cohorts of neonates undergoing therapeutic hypothermia and afterwards to generate such reference values."

We have adapted the last sentence of the introduction section, from

“We therefore conducted a structured search on published individual QTc intervals and QTc patterns in cohorts of neonates undergoing therapeutic hypothermia and afterwards to generate such reference values”

to

“We therefore conducted a structured search on published individual QTc intervals and QTc patterns in cohorts of neonates undergoing therapeutic hypothermia and afterwards during normothermia to construct a QTc pattern over postnatal age during and after therapeutic hypothermia”

I suggest the authors add to the discussion that other covariates (potassium, pH, disease severity) likely affect the QTc pattern. However, they had no access to such data data. Therefore, they could not address this point in the paper

We have further added this to the first alinea of the discussion: It is important to realize that the pattern of QTc intervals during and after therapeutic hypothermia is a phenotypic description, driven by hypothermia, but likely also in part determined by other covariates like potassium levels, pH, or disease severity.

Reviewer 2 Report

The authors have improved the manuscript. I have two further suggestions: 1- The inclusion and exclusion criteria should be better described; 2- insert a table summarizing the main findings of the selected studies. 

Author Response

The authors have improved the manuscript. I have two further suggestions: 1- The inclusion and exclusion criteria should be better described; 2- insert a table summarizing the main findings of the selected studies. 

Inclusion and exclusion criteria have been better described (search strategy, time, publication data and language were already mentioned. Inclusion: QTc data reported, in human neonates undergoing therapeutic hypothermia or shortly afterwards, exclusion: other topics, like unintentional hypothermia).

This journal has not a ‘bullet’ section with the main findings/key messages, and a Table is not commonly used to summarize main findings. We therefore felt that a summary at the end of the 4.1. findings would both respect the request of the reviewer, as well as the editing guidelines of the journal.

Based on 94 individual QTc observations in 33 neonates during of following therapeutic hypothermia (day 1-7), the median QTc was significantly prolonged (504 versus 410 ms) during therapeutic hypothermia (day 1-3) compared to subsequent normothermia (day 4-7). This difference is equal to -21.6 ms/°C, while this pattern constructed from the individual findings is similar to the pattern described in 4 cohorts.